# Targeting Solute Carrier Transporters (SLCs) as a Therapeutic Target in Different Cancers

**DOI:** 10.3390/diseases12030063

**Published:** 2024-03-21

**Authors:** Ravi Bharadwaj, Swati Jaiswal, Erandi E. Velarde de la Cruz, Ritesh P. Thakare

**Affiliations:** 1Department of Infectious Diseases, UMass Chan Medical School, Worcester, MA 01605, USA; 2Department of Molecular Cell and Cancer Biology, UMass Chan Medical School, Worcester, MA 01605, USA; erandi.velardedelacruz@umassmed.edu (E.E.V.d.l.C.); ritesh23thakare@gmail.com (R.P.T.)

**Keywords:** SLC, transporter, colon cancer, pancreatic cancer, liver cancer, lung cancer

## Abstract

Solute carrier (SLC) transporters constitute a vast superfamily of transmembrane proteins tasked with regulating the transport of various substances such as metabolites, nutrients, ions, and drugs across cellular membranes. SLC transporters exhibit coordinated expression patterns across normal tissues, suggesting a tightly regulated regulatory network governing normal cellular functions. These transporters are crucial for the transport of various metabolites, including carbohydrates, proteins, lipids, and nucleic acids. However, during tumor development, metabolic changes drive an increased demand for energy and nutrients. Consequently, tumor cells alter the expression of SLC transporters to meet their heightened nutrient requirements. Targeting SLCs through inhibition or activation presents a promising therapeutic approach in cancer treatment. Certain SLCs also serve as intriguing chemo-sensitizing targets, as modulating their activity can potentially alter the response to chemotherapy. This review underscores the significance of various SLCs in tumor progression and underscores their potential as both direct and indirect targets for cancer therapy.

## 1. Introduction

Transporters are specialized proteins that facilitate the movement of ions, peptides, small molecules, lipids, and macromolecules across biological membranes. These proteins play crucial roles in controlling nutrient levels, eliminating waste from cells, regulating cell volume, and maintaining cellular energy levels. For transporting a variety of compounds, specialized transporters have been classified into multiple types. However, the two major superfamilies of transporters are ABC transporters and solute carrier transporters, in addition to other transporter families [1,2]. ABC transporters utilize the energy from ATP hydrolysis and function as efflux transporters, whereas SLC transporters are primarily involved in the transporting of small molecules and ions from cells [3]. The present review provides an introductory overview of the SLC superfamily of transporter proteins and their crucial roles within human cells. The solute carrier family encompasses a diverse array of transporters that facilitate secondary active transport and facilitative diffusion mechanisms. These transporters are ubiquitously distributed across cellular compartments, spanning from the plasma membrane to the membranes of virtually every organelle. The expansive SLC superfamily comprises 65 families encompassing 458 transporters that facilitate the movement of diverse substances across cell membranes. These substances include amino acids, sugars, drugs, ions, and vitamins [4]. SLC transporters operate through secondary active transport or facilitative diffusion mechanisms, distinct from active transporters that directly utilize the energy from ATP hydrolysis. Within the SLC superfamily, transporters may function as symporters or antiporters, utilizing the electrochemical gradient of one substrate (typically ions) to provide the energy required for the transport of another substrate in an energetically unfavorable direction [5]. The proteins of the SLC superfamily have a variety of three-dimensional structures; not all but most SLC family members share some common features. According to the hydrophobicity plot, SLC proteins are predicted to contain 1 to 16 transmembrane (TM) domains. However, 83% of SLC proteins consist of 7 to 12 transmembrane domains [6,7]. So far, all the known structures of SLC proteins have been shown to have a unique feature: pseudosymmetry within their TM domains [8]. SLC proteins are classified based on various structural features. 

Among SLC proteins, two common structural folds are observed: the major facilitator superfamily (MFS) fold and the leucine transporter (LeuT-like fold. The MFS fold typically consists of two pseudo-repeats, each containing six transmembrane helices linked by a cytoplasmic loop. In contrast, the LeuT-like fold comprises two sets of five transmembrane helices, each featuring a bundle and scaffold domain. SLC proteins with the MFS fold utilize a rocker and switch mechanism, while those with the LeuT fold employ a rocking-bundle approach [8]. Additionally, some SLC proteins utilize a third mechanism known as the “elevator” mechanism [9,10]. In both the rocker-switch and the rocking-bundle models, substrate binding occurs at a binding site, initiating a shift in the transmembrane domains around the substrate. This exposes the binding site to the opposite side of the membrane, facilitating substrate release. Conversely, the elevator mechanism involves two distinct domains: a scaffold domain and a transport domain, each serving specific functions. In this mechanism, the transport domain binds the substrate and undergoes rigid body movement to translocate across the membrane, ultimately releasing the substrate [10]. In a recent article by Puris et al. and Lin et al. [11,12], the therapeutic potential of SLC transporters in cancer was explored. It was posited that the expression of SLCs is diminished in cancerous cells, potentially leading to compromised drug delivery [12], although this phenomenon remains relatively understudied. The present review seeks to consolidate existing knowledge regarding SLC transporter expression in various cancers compared to healthy tissues, as well as ongoing research endeavors aimed at elucidating the role of SLCs in cancer progression. Furthermore, recent advancements in strategies targeting SLC transporters, such as the development of transporter-utilizing prodrugs, and the modulation of SLC transporter expression in cancer cells, are discussed.

## 2. SLC Transporters in Cancer

Many studies have emphasized the crucial involvement of the SLC superfamily in various aspects of tumorigenesis, including invasion, proliferation, metastasis, apoptosis, chemotherapy resistance, and other cancer-related processes. SLC transporters facilitate the entry of diverse metabolically significant substrates into cells, which is essential for human physiology. These processes include glucose absorption, uptake of water-soluble vitamins crucial for vital functions, transportation of metal ions, amino acid intake in the intestine, and reabsorption of neurotransmitters into presynaptic neurons [11]. Tumorigenesis, a complex process, orchestrates the conversion of normal cells into highly malignant derivatives [12,13]. In certain cases, cancer cells may increase the expression of SLC transporters to fulfill nutritional needs, which gives them an advantage over normal cells when nutrients are scarce. For example, the phenomenon known as the Warburg effect comprises cancer cells increasing the uptake of glucose and aerobic glycolysis to meet the highly metabolically active state [14]. Conversely, certain SLC transporters can facilitate the delivery of specific drugs to cancer cells, presenting novel avenues to enhance chemotherapy sensitivity and counter drug resistance [15]. Notably, the SLC16 transporter family (figure not shown), also referred to as monocarboxylic acid transporters (MCTs), plays a pivotal role in cancer metabolism [15,16]. This family, comprising MCT1-4 (SLC16A1, SLC16A7, SLC16A8, SLC16A3), primarily mediates the transport of monocarboxylic acids, such as lactic acid, thereby regulating cellular pH and overall homeostasis [15,17]. In cancer cells, elevated glycolysis levels lead to upregulated expression of MCT1-4. Moreover, SLC16A1, SLC16A3, and SLC16A13 have emerged as promising biomarkers for prognostication in pancreatic cancer [18]. MCT5 (SLC16A4), expressed in the basolateral membrane of the colon, likely participates in nutrient absorption [19]. Lin et al. demonstrated upregulated MCT5 expression in colon cancer, underscoring its potential relevance in this context [20]. Moreover, MCT1 (SLC16A1) has been implicated in breast cancer, where its overexpression is frequently observed [21]. By selectively targeting MCT1, researchers have found that the disruption of pyruvate export significantly impacts the metabolic pathways essential for cancer cell proliferation. This approach effectively starves the cancer cells, leading to impaired growth and proliferation. These findings underscore the potential of MCT1 inhibition as a therapeutic strategy for targeting glycolytic breast cancer cells and highlight the importance of understanding the specific metabolic dependencies of cancer cells for developing targeted therapies [22]. The subsequent sections provide summaries of select SLC transporters implicated in various aspects of pancreatic tumor biology, elucidating their roles. We have delineated the functions of each differentially expressed SLC transporter in cancerous cells and their corresponding inhibitors in Table 1 and Figure 1, Figure 2, Figure 3, Figure 4, Figure 5 and Figure 6.

### 2.1. Pancreatic Cancer

Pancreatic cancer ranks as the seventh leading cause of cancer-related deaths worldwide, with approximately 432,242 new deaths annually according to GLOBOCAN 2018 [15,16]. Despite its relatively low incidence, pancreatic cancer remains highly lethal, boasting a dismal five-year overall survival rate of approximately 8%, solidifying its place as a significant contributor to global cancer mortality [18]. Within pancreatic cancer, ten SLC transporters have emerged as key players in chemoresistance, tumor proliferation, and, notably, tumor suppression (Figure 1).

One of these transporters, SLC1A5, encodes the alanine–serine–cysteine transporter 2 (ASCT2) protein responsible for transporting neutral amino acids in a sodium ion-dependent manner. SLC1A5 is predominantly expressed in mitochondria, facilitating the transport of glutamine and other neutral amino acids into these organelles [19,20]. Elevated expression of SLC1A5 has been observed in surgically extracted tumor samples compared to adjacent tissues or benign pancreatic lesions, indicating its potential role in pancreatic tumor progression. Knockdown experiments targeting SLC1A5 expression have resulted in a drastic reduction in tumor cell proliferation, further underscoring its significance in pancreatic cancer pathogenesis [19].

Similarly, SLC4A7, known as a sodium and bicarbonate ion cotransporter (NBC3), plays a pivotal role in regulating intracellular pH in pancreatic cells. Recent studies have implicated SLC4A7 in tumor growth by inducing micropinocytosis in SLC4A7 knockout Ras mutant pancreatic ductal adenocarcinoma (PDAC) cells [23].

SLC7A5, another transporter, regulates the influx of L-leucine and the efflux of L-glutamine. Its high expression has been correlated with CD147 expression, proliferation, angiogenesis, and mammalian target of rapamycin (mTOR) signaling in pancreatic cancer cells [24].

SLC7A11, a cystine/glutamate antiporter, promotes pancreatic cancer proliferation through its role in glutathione biosynthesis. Overexpression of SLC7A11 has been shown to enhance pancreatic cancer proliferation, while downregulation reverses this effect [25,26].

Members of the SLC30 and SLC39 families, which regulate zinc transportation with opposing functions, have also been implicated in pancreatic cancer. For instance, SLC39A4 exhibits significantly increased expression in pancreatic cancer tissues compared to normal tissue and has been linked to enhanced cell proliferation [27]. Similarly, SLC39A6, a downstream target of STAT3, plays a crucial role in regulating the epithelial-to-mesenchymal transition (EMT) in pancreatic cancer cells [28].

SLC5A8, a sodium-coupled monocarboxylate transporter 1 (SMCT1), transports monocarboxylates such as lactate, butyrate, pyruvate, acetate, propionate, nicotinate, and β-hydroxybutyrate. Its reduced expression and nuclear translocation have been associated with the Warburg effect in tumor cells and poor prognosis in pancreatic cancer patients [29,30,31,32,33,34].

Additionally, SLC39A3 and SLC41A1 have been reported to suppress tumor growth by downregulating Ras-responsive element-binding protein 1 (RREB1) and suppressing the activation of Akt/mTOR signaling, respectively [35,36,37]. These findings collectively highlight the diverse roles of SLC transporters in pancreatic cancer progression and underscore their potential as therapeutic targets for this deadly disease.

### 2.2. Breast Cancer

Breast cancer (BC) stands as the most prevalent malignancy in females and remains a leading cause of cancer-related mortality worldwide, constituting 12% of new cases annually in the United States [38]. Although the role of ABC transports in BC has been extensively studied, recent attention has shifted towards other membrane transporters, particularly members of the SLC family, which are pivotal in transporting a diverse range of metabolites and ions across cellular membranes (Figure 2) [4]. Among these, SLCO1A2 (OATP1A2) emerges as a significant player, being expressed in various tissue types and primarily involved in transporting prostaglandins, steroid hormones/conjugates, and anticancer drugs [39]. BC cells exhibit a tenfold increase in SLCO1A2 expression compared to healthy tissue, facilitating the uptake of dehydroepiandrosterone sulfate (DHEA-S) and potentially promoting BC cell proliferation. Inhibition of SLCO1A2 activity restricts BC cell growth [40,41,42]. High expression of SLCO1B1 (OATP1B1) and SLCO1B3 (OATP1B3) has also been observed in BC, correlating with increased uptake rates of paclitaxel and potentially influencing the deposition of clinically relevant drugs. Notably, SLCO1B3 expression inversely correlates with tumor size and is associated with a decreased risk of recurrence [43,44,45].

Another noteworthy SLC transporter in BC is SLCO2A1 (OATP2A1), identified as a prostaglandin transporter, which is upregulated in BC cell lines and tissues compared to normal tissue. Elevated expression of SLCO2A1 has been linked to increased intra-tumoral prostaglandins, contributing to several cancer hallmarks and supporting a cancer stem cell-like phenotype [46,47]. SLCO3A1 and SLCO4A1, widely expressed transporters, exhibit relatively high expression in BC cell lines and malignant tissues, playing essential roles in E-3-S transport and BC proliferation [48,49]. Both SLCO4A1 and SLCO4C1 facilitate the excretion of various drugs and compounds, including methotrexate, thyroid hormones, and E-3-S, with elevated expression reported in BC cell lines [49,50,51,52,53]. A comprehensive analysis of SLC protein expression and function in BC is crucial to elucidate their role in disease progression. Functional experiments can provide insights into the interplay between SLC transporters, nuclear receptors, and ABC transporters, potentially identifying new targets to impede BC progression to metastatic stages.

### 2.3. Leukemia

Leukemia, a progressive malignancy affecting blood-forming organs, is characterized by abnormal proliferation and development of leukocytes in the blood and bone marrow. In the United States, approximately 29,000 adults and 2000 children are diagnosed with leukemia annually. Leukemia is classified into lymphocytic and myeloid types based on the involved progenitor cells [54]. Major solute carrier (SLC) proteins serve as therapeutic targets in leukemia (Table 1, Figure 3).

**Table 1 diseases-12-00063-t001:** List of SLC transporters expressed in different tumors, their function, and relevant inhibitors.

Cancer	Solute Carrier Transporters	Function	Inhibitors/Substrates
**Pancreatic cancer**	SLC1A5 (alanine–serine–cysteine transporter 2)	Transports neutral amino acid with co-transportation in a sodium ion-dependent manner	V9302 [55]
	SLC4A7 (sodium and bicarbonate ion co-transporter)	Plays a critical role in regulating the intracellular pH of pancreatic cells	S0859 [23]
	SLC7A5 (L-type amino acid transporter 1; LAT1)	Regulates the influx of L-leucine and the efflux of L-glutamine	JPH203 [56], KYT 0353 [57,58], melphalan, acivicin [11]
	SLC7A11 cystine/glutamate antiporter)	Raises glutathione biosynthesis through the uptake of cystine and the release of glutamate [22]	HG106 [11,59]
	SLC30	Involved in the efflux of zinc	-
	SLC39A3, SLC39A4, and SLC39A6	Involved in the absorption of zinc and increased cell proliferation	-
	SLC5A8 (Na^2+^-coupled monocarboxylate transporter 1	Co-transports monocarboxylates such as lactate, butyrate, pyruvate, acetate, propionate, nicotinate, and β-hydroxybutyrate [29,30,31]	-
	SLC41A1	Suppress tumor growth [33,34,35]	-
	SLC22A5 (OCTN2)	Zwitterions (L-carnitine), organic cations	Etoposide, imatinib [11]
**Breast Cancer**	SLCO1A2 (OATP1A2)	Involved in transporting prostaglandin and steroid hormones/conjugates; estradiol-17β-glucuronide, estrone-3-sulfate (E-3-S), dehydroepiandrosterone sulfate (DHEA-S), and anticancer drugs; and imatinib, methotrexate, paclitaxel, doxorubicin, and docetaxel [39]	Imatinib, methotrexate [11]
	SLCO1B1 (OATP1B1) and SLCO1B3 (OATP1B3)	Related to drug uptake	Cisplatin, carboplatin, oxaliplatin, regorafenib, belzutifan, SN-38, etoposide, tamoxifen, nilotinib, docetaxel, imatinib, gefitinib [11], sorafenib [60]
	SLCO2A1 (OATP2A1) and SLCO2A4	Involved in prostaglandin transport	Suramin [61]
	SLCO4A1 and SLCO4C1	Promote the elimination of external medications and internal substances such as methotrexate, thyroid hormones, and E-3-S [48,49]	-
	SLC22A5 (OCTN2)	Transports zwitterions (L-carnitine), organic cations	Etoposide, imatinib [11]
**Leukemia**	SLC38A1	Transports glutamine and plays a crucial role in maintaining the homeostasis of the human body [62]	-
	SLC2 family (SLC2A5, SLC2A10, and SLC2A13)	Encodes glucose transporter (GLUT) proteins [63]	MSNBA and H22954 [64]
	SLC19A1	Transports reduced folate and found to be associated with methotrexate drug	Methotrexate, pemetrexed [11]
	SLC29A1 (ENT1)	Cellular uptake of anticancer nucleoside agents as well as physiologic nucleosides	Gemcitabine, cytarabine, 5-fluorouracil, 6-mercaptopurine [11], NBMPR [65]
	SLC22A1 (OCT1)	Transport of organic cations, i.e., nutrients, neurotransmitters, metabolites, or drugs [66]	Nintendanib [11]
	SLC1A5	Uptake of neutral amino acids, including glutamine (Gln), cysteine (Cys), serine (Ser), threonine (Thr), valine (Val), and alanine (Ala) [67,68]	V9302 [55] GPNA [69]
	SLC7A7	Transports cationic amino acids such as arginine and lysine out of the cell	-
**Colon cancer**	SLC19A3 (thiamine transporter 2; THTR2)	Manipulates the DNA methylation and histone deacetylation status of the nucleic acid of these cells	-
	SLC3A2	Putrescine and arginine importer	-
	SLCO4A1	Transport of various compounds, including sugars, bile salts, organic acids, metal ions, amine compounds, and estrogen, and high expression has been reported in many cancer types, including colon cancer [70,71]	-
	SLC22A3 (OCT3)	Organic cation transporter 3	Oxaliplatin [11]
	SLC35A2 and SLC29A1	Nucleoside transporter	-
	SLC1A5	Glutamine transporter	Cetuximab [72]
	SLC10A2	Associated with the progression of colorectal cancer, especially in kids [73]	-
	SLC7A2	Inducible transporter of the semi-essential amino acid L-arginine (L-Arg)	-
	SLC37A1	Involved in the sugar–phosphate exchange	-
**Lung cancer**	SLC18A1	Transports monoamines, such as norepinephrine, epinephrine, dopamine, and serotonin, and has been associated with neuropsychiatric disorders [74]	-
	SLC39A3, SLC39A4, and SLC39A7	Zn^2+^ influx transporter, essential role in cell survival of lung adenocarcinoma	NVS-ZP7-4 [69]
	SLC1A5	Transports glutamine and regulates the cell growth and oxidative stress in non-small cell lung cancer [75]	Gamma-l-glutamyl-p-nitroanilide (GPNA) [76]
	SLC29A3	Nucleoside transporter	NBMPR [65]
	SLC7A5	Plays important role in non-small cell lung cancer [77]	JPH203 [56], KYT 0353 [58], melphalan, acivicin [11]
	SLCO2A1 (OATP2A1)	Organic anion transporter, reported to function in the progression of lung cancer	Suramin [61]
	SLC38A3	An amino acid transporter, specifically of L-glutamine, L -histidine, L -alanine, and L -asparagine	-
	SLC22A16 (OCT6)	Mediates platinum influx into cancer cells	-
	SLC22A18	Regulates cellular metabolism, cellular growth, and drug sensitivity [78]	-
	SLC22A5 (OCTN2)	Zwitterions (L-carnitine), organic cations	Etoposide, imatinib [11]
**Liver cancer**	SLC39A6	Significantly upregulated in liver cancer tissues compared with normal liver tissues [79]	-
	SLC26A6	Nonselective anion exchanger that transports anions such as oxalate, formate, and sulfate [80]	DIDS [81]
	SLC2A1	Glucose transporter and increased in HCC	GRg3, 1:1 CC-DG mixture [64]
	SLC2A2	Glucose transporter and decreased in HCC [82,83]	Streptozotocin [11]
	SLC13A5	A sodium-coupled citrate transporter, plays a key role in importing citrate from the circulation into liver cells [84,85]	BI01383298 [86]
	SLCO2B1 (OATP2B1)	An organic anion transporter; decreased expression in liver cancer	Etoposide, erlotinib [11]
	SLCO2A1 (OATP2A1)	Increased level of mRNA observed in hepatic tumors	Suramin [61]

Solute carrier family 38 member 1 (SLC38A1), primarily transporting glutamine, is crucial for maintaining human body homeostasis. Dysregulation of SLC38 transporters may contribute to tumor initiation and progression, with high SLC38A1 expression linked to poorer prognosis in acute myeloid leukemia (AML) and shorter overall survival [87].

Members of the SLC2 family, including SLC2A5, SLC2A10, and SLC2A13, are associated with AML prognosis. High SLC2A5 and SLC2A10 expression correlates with poor survival, while low SLC2A13 expression is linked to increased mortality [63,88].

Solute carrier family 19 member 1 (SLC19A1) transports reduced folate and is involved in methotrexate drug transportation in leukemia. Higher SLC19A1 expression predicts better overall survival [89].

Solute carrier family 29 member 1 (SLC29A1 or ENT1) facilitates cellular uptake of anticancer nucleoside agents. Elevated SLC29A1 expression disrupts 5-fluorouracil (5-FU) absorbance, impacting cancer treatment [61].

Organic cation transporter 1 (OCT1), encoded by SLC22A1, transports organic cations. Reduced OCT1 expression in AML patients correlates with higher chances of achieving complete remission and longer overall survival [60,66].

SLC1A5, a transmembrane transporter for neutral amino acids, is highly expressed in AML. Inhibition disrupts leucine influx, induces apoptosis, and impedes leukemia progression [67,68,90].

Amino acid transporter SLC7A7 influences T-cell acute lymphoblastic leukemia pathogenesis by enhancing arginine transport in cancer cells [91,92,93,94].

### 2.4. Colon Cancer

Colon cancer, the third most common and fourth deadliest cancer globally, claims almost 900,000 lives annually and predominates as the most prevalent gastrointestinal cancer [94]. Mika et al. demonstrated in colon cancer cell lines that altering the expression of thiamine transporter 2 (THTR2) (SLC19A3) can modulate DNA methylation and histone deacetylation, associated with thiamine deficiency in advanced colon cancer. Glutamine transporter solute carrier family 1 member 5 (SLC1A5) inhibition or downregulation in SW620/Ad300 cells reduces glutamine supply, impeding cancer cell proliferation [95]. OCT1 transporter (SLC22A1) mRNA expression increases in colon cancer cell lines and patient-derived colorectal tumor samples, proposed as a prognostic marker for colon cancer [96,97]. SLCO1B3 (OATP1B3; not shown in the figure) overexpression in HCT116 colon cancer cells inhibits apoptosis and enhances cell survival by suppressing p53 activity through SN38 [98]. Knockdown of SLC3A2, an importer of putrescine and arginine, triggers DNA replication decrease and autophagy induction in colon cancer cells [99,100]. High expression of SLCO4A1, implicated in transporting various compounds, including estrogen, in colon cancer enhances carcinogenic abilities and E1S uptake, impacting colorectal cancer prognosis [70,71,101]. High levels of nucleoside transporter SLC29A1 correlate with poor clinical response to 5-FU in colorectal tumor tissue [102], while SLC35A2 is abundantly expressed in colon cancer cells compared to non-malignant tissue. SLC22A3 expression significantly increases in colorectal cancer tissues, with its inhibition in colon cancer cell lines suppressing cancer cell proliferation, migration, and invasion and promoting apoptosis [103]. SLC5A8 transports the bacterial fermentation product butyrate, acts as a conditional tumor suppressor in colon cancer, and is linked to dietary fiber content [73]. Dysfunction in bile acid transporter ASBT (SLC10A2) is associated with colorectal cancer advancement, particularly in pediatric cases [73], and SLC7A2 deficiency exacerbates colitis and increases the risk of colitis-associated colon tumorigenesis [104]. SLC37A1 expression significantly increases in colon tumor tissues, which is associated with hematogenous metastasis and glycolipid metabolism, indicating a potential role in colon cancer progression [105]. High GLUT1 staining correlates with an increased risk of death from colon carcinoma [106].

### 2.5. Lung Cancer

Lung cancer ranks among the most prevalent and deadliest cancers globally, with approximately 2 million new cases and 1.76 million deaths annually [107].

SLC18A1, a vesicular monoamine transporter facilitating the transport of monoamines like norepinephrine, epinephrine, dopamine, and serotonin, has been linked to neuropsychiatric disorders. Alterations in SLC18A1 expression have been observed in lung cancer patients, and its function is speculated to complement other biomarkers under study in PD1/PD-L1 inhibition [74,108].

The SLC39A family, acting as Zn2+ influx transporters, has several members associated with different lung cancer forms. SLC39A3, SLC39A4, and SLC39A7 expressions are linked to lung squamous cell carcinoma prognosis, with SLC39A7 playing a crucial role in lung adenocarcinoma (LUAD) cell survival and proliferation [109,110].

Solute carrier family A1 member 5 (SLC1A5) is primarily involved in glutamine transport, regulating cell growth and oxidative stress in non-small cell lung cancer (NSCLC). Inhibition or knockdown of SLC1A5 decreases glutamine content, reducing cell growth and viability through mTOR signaling [75,79].

SLC29A3, encoding ENT3, affects overall survival (OS) and response to gemcitabine-based chemotherapy in lung cancer patients. Elevated OCTN1 and OCTN2 mRNA levels are reported in human lung carcinoma, while LAT1 (SLC7A5) is upregulated in NSCLC [77,111,112,113].

SLCO2A1, an organic anion transporter, facilitates prostaglandin E2 uptake, promoting lung cancer cell invasion via the PI3K/AKT/mTOR pathway [114,115].

SLC38A3 upregulation in metastatic NSCLC cells correlates with patient prognosis, promoting metastasis through PDK1/AKT signaling activation by reducing glutamine and histidine concentrations [116].

SLC22A16 (OCT6), a high-affinity carnitine transporter, mediates platinum influx into cancer cells, and its downregulation contributes to acquired platinum resistance in lung cancer. SLC22A18 aberrant expression is observed in various cancers, including NSCLC, impacting cellular metabolism, growth, and drug sensitivity [78,113,117,118,119]. However, the molecular mechanisms underlying SLC22A18’s roles in NSCLC remain unclear.

### 2.6. Liver Cancer

Liver cancer ranks among the most prevalent malignant solid tumors worldwide, standing as the fourth leading cause of cancer-related mortality and sixth in cancer incidence, with projections indicating more than one million deaths by 2030 [120]. Hepatocellular carcinoma (HCC) accounts for 70–85% of primary liver cancer cases [121]. Chronic liver diseases, including hepatitis B and C infections, non-alcoholic fatty liver disease, and alcohol abuse, are frequent precursors to liver cancer development [122,123,124]. Recent investigations have highlighted a significant upregulation of SLC39A6 in liver cancer tissues compared to normal liver tissues, correlating with poorer overall survival rates [125]. Solute carrier family 26 member 6 (SLC26A6) exhibits overexpression in human HCC and is associated with various cancer-related pathways [80,126]. Human organic solute carrier protein 1 (hOSCP1) variants predispose individuals to non-viral liver carcinoma [126]. Glucose transporters SLC2A1 and SLC2A2 are implicated in HCC; SLC2A1 is elevated, while SLC2A2 is decreased, correlating with clinical characteristics and prognosis [82,83,127]. SLC13A5, a sodium-coupled citrate transporter, plays a crucial role in hepatic energy homeostasis and hepatoma cell proliferation [84,85]. Organic anion-transporting polypeptides (OATPs) are upregulated in various solid tumors, including liver cancer, with SLCO2A1 (OATP2A1) elevated in HCC and liver metastases from colon cancer, while SLCO1B1 expression decreases in liver cancer, showing a reverse trend with cancer grade [128,129]. There is an increased expression of SLC46A3 reported in HCC patients’ livers [12]; however, we found that it significantly expresses in the gut region and plays an important role in bacterial muropeptide transportation [57,130].

## 3. Conclusions and Future Directions

Transporters play an important role in the modulation of metabolism and ion homeostasis of tumor cells, both important factors in proliferation, apoptosis, hypoxia, and migration [15]. SLC transporters represent a plethora of new therapeutic targets for rare diseases and many metabolic disorders by being amenable to targeting small molecules [11,17]. Importantly, many SLC transporters are expressed on the cell surface and therefore could be targeted by small molecules and therapeutic antibodies [15,131]. Additionally, numerous SLC transporters serve as targets for both already-approved medications and drugs currently in the developmental stage. Furthermore, extensive research is imperative to unveil potential variances in transporter expression among various cancer subtypes, particularly considering that cancers induced by mutations in oncogenes or tumor suppressor genes may exhibit substantial differences in their expression of drug transporters. Such investigations would offer insights into the regulation of transporter expression across different cancer subtypes and could provide valuable hints on selectively modulating drug transporter expression to enhance the delivery of anticancer drugs. Uncontrolled cell proliferation is a fundamental trait of cancer cells. Recent studies suggest that SLC transporters exhibit differential expression between proliferating and migrating, invasive tumor cells. Although some of these transporters may be regulated by the tumor microenvironment, much remains unknown about the relevance of distinct tumor cell subpopulations in tumor progression, dissemination, variations in transporter expression among these subpopulations, and the impact of surrounding tissue on transporter regulation. Transporters offer various avenues for exploitation in cancer treatment, including direct targeting using small molecules or antibodies to inhibit their function, or utilizing them as carriers for drugs to enhance drug efficacy. Despite recent strides in investigating SLC transporters, significant gaps persist in understanding their contribution to anticancer drug delivery and the diminished drug efficacy resulting from low expression of drug influx transporters. Nevertheless, promising innovative strategies proposed for augmenting transporter-mediated anticancer drug delivery hold the potential for overcoming transporter-mediated anticancer drug resistance and advancing the development of efficacious anticancer treatments.

## Figures and Tables

**Figure 1 diseases-12-00063-f001:**
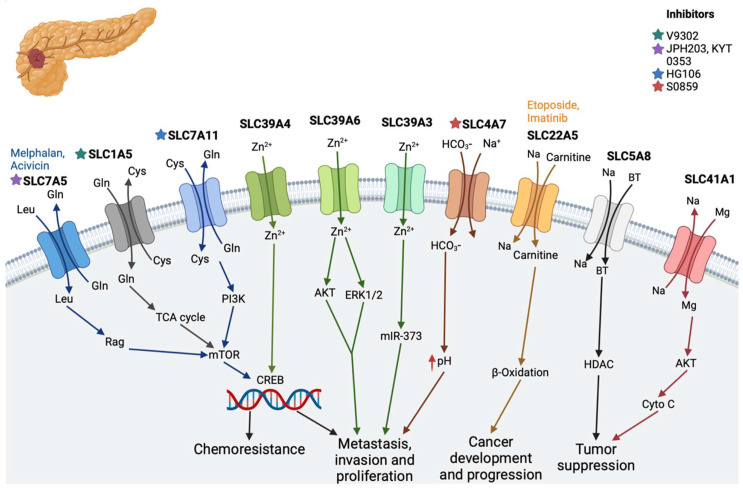
SLC transporters in pancreatic cancer. SLC7A5, SLC1A5, and SLC7A11 help with amino acid transport and play a role in chemoresistance and metastasis by activating the mTOR pathway. SLC39A4 activates the CREB transcription factor and plays a role in metastasis, invasion, and cell proliferation together with SLC39A6 and SLC39A3. SLC4A7 transports sodium bicarbonate and helps with tumor proliferation. SLC22A5 helps with tumor progression, while SLC5A8 and SLC41A1 play a role in tumor suppression. (Stars show the inhibitors of specific transporters and the substrates are in the colored text.) The image was created in BioRender.

**Figure 2 diseases-12-00063-f002:**
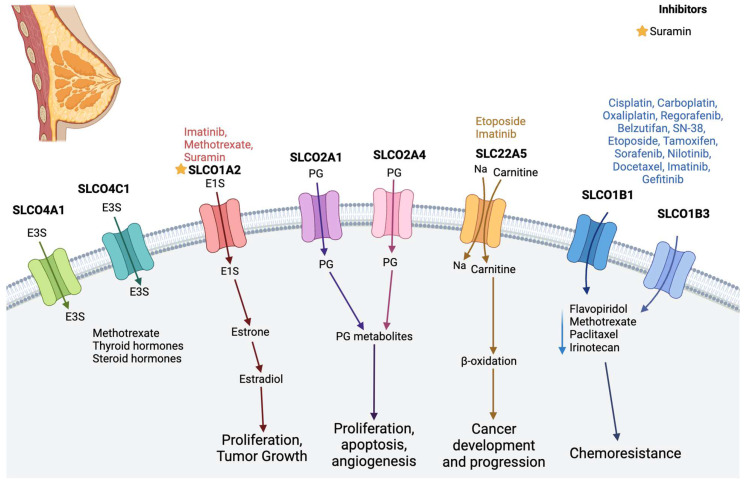
SLC transporters in breast cancer. SLCO4A1, SLCO4C1, and SLCO1A2 transport steroid hormones (E3) and play a role in cell proliferation and tumor growth. SLCO1B1 and SLCO1B3 promote chemoresistance, as they transport several chemotherapy drugs. SLCO2A1 and SLCO2A4 transport progesterone (PG) and promote angiogenesis and proliferation. SLC22A5 helps with cancer development (see text). The image was created in BioRender.

**Figure 3 diseases-12-00063-f003:**
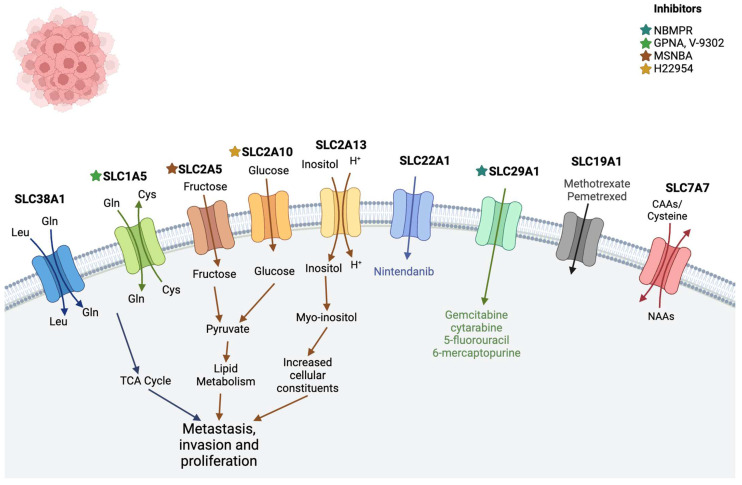
SLC transporters in leukemia. The SLC38A1, SLC1A5, and SLC2 families promote metastasis, invasion, and cell proliferation (see text). Other transporters like SLC22A1, SLC29A1, SLC19A1, and SLCA7 play important roles in transporting anti-cancer drugs. (Stars show the inhibitors of specific transporters and the substrates are in the colored text.) The image was created in BioRender.

**Figure 4 diseases-12-00063-f004:**
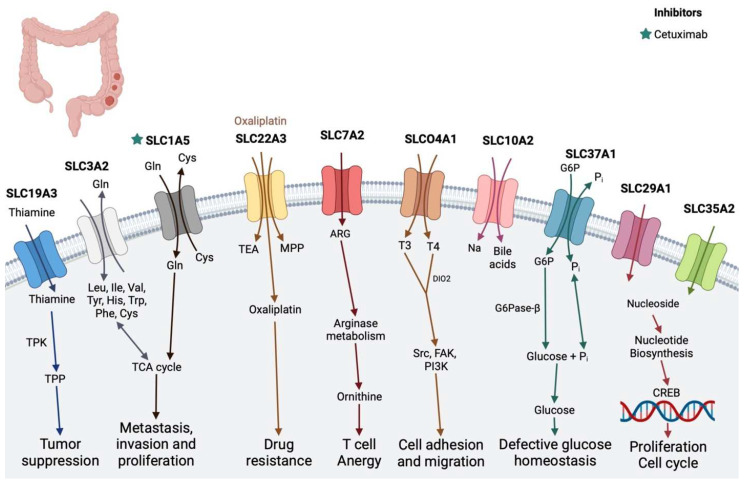
SLC transporters in colon cancer. SLC7A5, SLC1A5, and SLC39A4 play a role in chemoresistance by activating the CREB transcription factor. SLC39A7 plays a role in metastasis. (Stars show the inhibitors of specific transporters and the substrates are in the colored text.) The image was created in BioRender.

**Figure 5 diseases-12-00063-f005:**
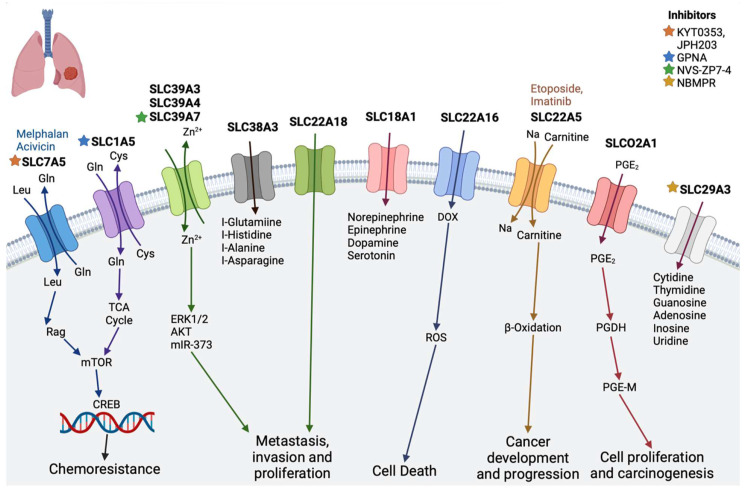
SLC transporters in lung cancer. SLC7A5, SLC1A5, SLC39A3, SLC39A4, and SLC39A7 promote chemoresistance as in pancreatic cancer. SLC22A18 plays a role in metastasis. SLC18A1, SLC29A3, and SLC38A3 transport amino acids and monoamines (see text). SLC22A16 transports doxorubicin (DOX). (Stars show the inhibitors of specific transporters and the substrates are in the colored text.) The image was created in BioRender.

**Figure 6 diseases-12-00063-f006:**
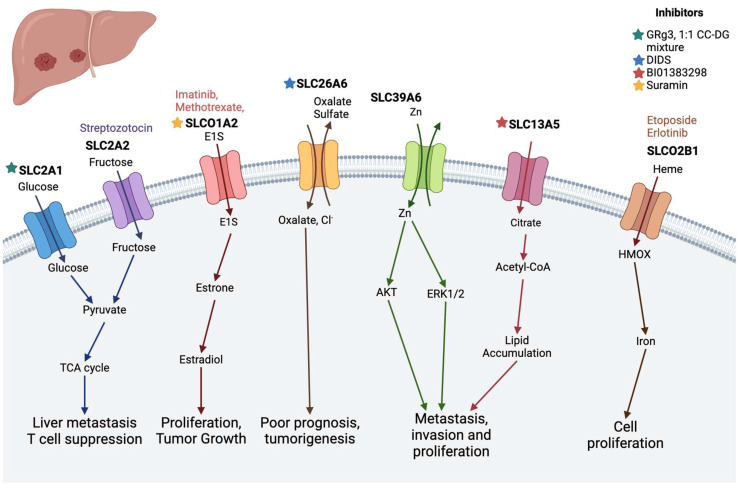
SLC transporters in liver cancer. SLC2A1 and SLC2A2 transport glucose and fructose, respectively, and promote metastasis. SLC26A6 plays a role in tumorigenesis. SLC39A6, SLC13A5, SLCO2A1, and SLCO2B1 are important in cell proliferation. (Stars show the inhibitors of specific transporters and the substrates are in the colored text.) The image was created in BioRender.

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
