# Peer review of "Targeting Solute Carrier Transporters (SLCs) as a Therapeutic Target in Different Cancers"

_diseases, 2024, doi:10.3390/diseases12030063_

Round 1

Reviewer 1 Report (Previous Reviewer 1)

Comments and Suggestions for Authors

The authors have addressed my concerns regarding the manuscript thus my decision is to accept it for publication.

Comments on the Quality of English Language

No major concerns on this topic.

Author Response

We are grateful for the reviewer's endorsement. 

Reviewer 2 Report (Previous Reviewer 4)

Comments and Suggestions for Authors

Although several paragraphs were changed in the revised version, modifications of several sections are still necessary to improve the overall quality and scientific merit of the paper according to my opinion. The paper could also benefit from a thorough re-checking of all statements and the related references. My previous comments were appropriately addressed in the Authors’ responses, however, several suggested corrections were not done in the text.

General comment: Please discuss Warburg effect, the complex metabolic change occurring upon carcinogenesis. Since SLC transporters are involved in cellular uptake of nutrients and export of metabolic side-products, changes of cellular metabolism should entail changes of membrane transport mechanisms such as changes in the expression pattern of transporters.

Specific comments:

Rows 28-29: These statements contain several over-simplifications. Please consider that besides ABC and SLC transporters there are several other classes of membrane transporters (e.g., P-type, V-type, F-type ATPases, please see Cell Biology textbooks).

Rows 30-32: Please do not forget about the fact that SLC transporters are also involved in efflux of molecules/ions from cells.

Rows 40-42: Please consider that transporters directly using the energy of ATP hydrolysis for transporting substances through the membrane called primary active transporters.

In the sentence ending in row 91 please cite the following recently published review: Molecules 2023 Jan 24;28(3):1151. doi: 10.3390/molecules28031151.

Rows 91-96: More citations are needed, such as DOI: 10.1073/pnas.2022495118 and Cell Rep. 25, 3047–3058.e4 (2018).

Rows 104-105: This statement is not clear, probably not true.

Row 231: Please insert appropriate references that really support the role of OCT1 in chemotherapy sensitivity of AML, such as Oncotarget. 2018; 9:28474-28485. https://doi.org/10.18632/oncotarget.25494

Rows 339-341: References are needed, such as DOI: 10.3390/molecules28031151

Rows 341-344: References are needed, such as doi: 10.1073/pnas.2022495118

Figure 3: Please modify OCT-1 substrates on this Figure, since according to publications dasatinib is not a substrate of OCT-1. „Dasatinib cellular uptake is not significantly affected by OCT-1 activity, so that expression and function of OCT-1 is unlikely to affect response to dasatinib. Dasatinib is a substrate of both efflux proteins, ABCB1 and ABCG2.” (Clin Cancer Res (2008) 14 (12): 3881–3888.;)

Giannoudis A, et al. Effective dasatinib uptake may occur without human organic cation transporter 1 (hOCT1): implications for the treatment of imatinib-resistant chronic myeloid leukemia. Blood, 2008: 112:8 3348-3354.

Rows 235-236. References are not sufficient to support the role of SLC7A7 in the pathogenesis of T-cell acute lymphoblastic leukemia.  Please cite the following papers: Grzes, K., Swamy, M., Hukelmann, J. et al. Control of amino acid transport coordinates metabolic reprogramming in T-cell malignancy. Leukemia 31, 2771–2779 (2017). https://doi.org/10.1038/leu.2017.160  and Cellular Physiology and Biochemistry (2018) 48 (2): 731–740. https://doi.org/10.1159/000491899

Comments on the Quality of English Language

Instead of hydrophobic plot please use hydrophobicity profile/hydrophobicity plot or hydropathy plot.

Author Response

Comments and Suggestions for Authors

Although several paragraphs were changed in the revised version, modifications of several sections are still necessary to improve the overall quality and scientific merit of the paper according to my opinion. The paper could also benefit from a thorough re-checking of all statements and the related references. My previous comments were appropriately addressed in the Authors’ responses, however, several suggested corrections were not done in the text.

Author's response: We are grateful for the reviewer's endorsement and have implemented the suggested modifications as per their recommendations.

General comment: Please discuss Warburg effect, the complex metabolic change occurring upon carcinogenesis. Since SLC transporters are involved in cellular uptake of nutrients and export of metabolic side-products, changes of cellular metabolism should entail changes of membrane transport mechanisms such as changes in the expression pattern of transporters.

Author's response: We have incorporated a discussion on the Warburg effect into the introduction section and integrated it into the main text as per the reviewer’s suggestion.

Specific comments:

  1. Rows 28-29: These statements contain several oversimplifications. Please consider that besides ABC and SLC transporters there are several other classes of membrane transporters (e.g., P-type, V-type, F-type ATPases, please see Cell Biology textbooks).

Author's response: We have made modifications to the text to improve its clarity and language.

  1. Rows 30-32: Please do not forget about the fact that SLC transporters are also involved in the efflux of molecules/ions from cells.

Author's response: We thank the reviews for highlighting this error. We have modified the text.

  1. Rows 40-42: Please consider that transporters directly using the energy of ATP hydrolysis for transporting substances through the membrane called primary active transporters.

Author's response: We have revised the text to enhance clarity and language.

  1. In the sentence ending in row 91 please cite the following recently published review: Molecules 2023 Jan 24;28(3):1151. doi: 10.3390/molecules28031151.

Author's response: We have included the suggested citation.

  1. Rows 91-96: More citations are needed, such as DOI: 10.1073/pnas.2022495118 and Cell Rep. 25, 3047–3058.e4 (2018).

Author's response: We have incorporated the recommended citations into the text.

  1. Rows 104-105: This statement is not clear, probably not true.

Author's response: We have revised the sentence to make it assertive. 

  1. Row 231: Please insert appropriate references that really support the role of OCT1 in chemotherapy sensitivity of AML, such as  2018; 9:28474-28485. https://doi.org/10.18632/oncotarget.25494

Author's response: We have incorporated the recommended citations into the text.

  1. Rows 339-341: References are needed, such as DOI: 10.3390/molecules28031151

Author's response: We have incorporated the recommended citations into the text.

  1. Rows 341-344: References are needed, such as doi: 10.1073/pnas.2022495118

Author's response: We have incorporated the recommended citations into the text.

  1. Figure 3: Please modify OCT-1 substrates on this Figure, since according to publications dasatinib is not a substrate of OCT-1. „Dasatinib cellular uptake is not significantly affected by OCT-1 activity, so that expression and function of OCT-1 is unlikely to affect response to dasatinib. Dasatinib is a substrate of both efflux proteins, ABCB1 and ABCG2.” (Clin Cancer Res(2008) 14 (12): 3881–3888.;)

Author's response: We thank the reviewer for bringing attention to the issue with the figure. We have revised the figure as per the suggested modifications.

  1. Giannoudis A, et al. Effective dasatinib uptake may occur without human organic cation transporter 1 (hOCT1): implications for the treatment of imatinib-resistant chronic myeloid leukemia. Blood, 2008: 112:8 3348-3354. –

Author's response: We appreciate the reviewer's feedback and have revised both the figure and table according to their suggestions.

  1. Rows 235-236. References are not sufficient to support the role of SLC7A7 in the pathogenesis of T-cell acute lymphoblastic leukemia.  Please cite the following papers: Grzes, K., Swamy, M., Hukelmann, J. et al.Control of amino acid transport coordinates metabolic reprogramming in T-cell malignancy. Leukemia 31, 2771–2779 (2017). https://doi.org/10.1038/leu.2017.160  and Cellular Physiology and Biochemistry (2018) 48 (2): 731–740. https://doi.org/10.1159/000491899

Author's response: We have incorporated the recommended citations into the text.

Comments on the Quality of English Language

  1. Instead of hydrophobic plot please use hydrophobicity profile/hydrophobicity plot or hydropathy plot.

Author's response: We are grateful for the reviewer's input and have accordingly updated the plots as per their suggestions.

Reviewer 3 Report (Previous Reviewer 3)

Comments and Suggestions for Authors

Author's response: It is crucial to emphasize that while the names of certain transporters may indeed be lengthy and bear resemblance across multiple articles, their fundamental roles remain consistent. Hence, the similarity in terminology does not inherently imply plagiarism.

It is important to acknowledge that plagiarism detection software, while valuable, possesses its limitations. Despite our conscientious efforts to adhere to ethical writing practices, there may still be instances where similarities are flagged.

My comments: I disagree with the response to reviewers regarding the plagiarism comment. I have sent you the plag report from ithenticate, which is a well established and universally accepted software for plagiarism detection. The authors claim that only certain terminologies were kept as such from the source article which has caused the plagiarism. I defer to this comment. I request the editors to check the plagiarism checked file and understand that 17 % has been copied from a single source in which very few terminologies are present. The maximum sentences were copied in verbatim. I suggest the authors to response to reviewer's comments in a proper manner in a professional manner.

Regarding the revised submission, the authors have corrected the manuscript and the revised version has very less plagiarism also. The grammatical mistakes have been addressed.

So, I accept the article in its present form for publication.

Author Response

Thanks for taking the time to review our revised manuscript. We are grateful for the reviewer's endorsement.

This manuscript is a resubmission of an earlier submission. The following is a list of the peer review reports and author responses from that submission.

Round 1

Reviewer 1 Report

Comments and Suggestions for Authors

The review paper about the SLC superfamily in the context of cancer is a very pertinent subject. Furthermore, this review covers different classes of SLCs and targets different types of cancer. My decision is to accept the publication of this manuscript but I still have some comments that should be attended:

- In my opinion, it is missing at least an important family of SLC transporters which is the SLC16 which codifies for the monocarboxylate transporters. Some members of this family are overexpressed in different cancers. Information regarding this family should be added.

- Some minor formatting and language issues were detected. A thorough proofreading must be conducted

Comments on the Quality of English Language

The quality of English is good. Just some minor errors on language to be corrected.

Author Response

Reviewer 1.

The review paper about the SLC superfamily in the context of cancer is a very pertinent subject. Furthermore, this review covers different classes of SLCs and targets different types of cancer. My decision is to accept the publication of this manuscript but I still have some comments that should be attended:

- In my opinion, it is missing at least an important family of SLC transporters which is the SLC16 which codifies for the monocarboxylate transporters. Some members of this family are overexpressed in different cancers. Information regarding this family should be added.

Author’s Response: Thank you for your suggestion. Your feedback serves as invaluable guidance for enhancing the quality and clarity of our manuscript. We are fully committed to addressing the identified issues and ensuring that the review achieves a higher level of coherence and accuracy. We have included the SLC16 family transporter in the manuscript.

- Some minor formatting and language issues were detected. A thorough proofreading must be conducted

Author’s Response: We have revised the manuscript thoroughly.

Reviewer 2 Report

Comments and Suggestions for Authors

In the present manuscript, authors have explored " Targeting Solute Carrier Transporters (SLCs) as a therapeutic target in different Cancers". The study lacks a specific hypothesis being tested.

·         Typo errors and grammatical error to be rectified

·         Reference should be aligned as per Journal instruction

Comments on the Quality of English Language

English looks fine.

Author Response

Reviewer 2.

In the present manuscript, the authors have explored " Targeting Solute Carrier Transporters (SLCs) as a therapeutic target in different Cancers". The study lacks a specific hypothesis being tested.

  • Typo errors and grammatical errors to be rectified
  • Reference should be aligned as per Journal instruction

Author’s Response: Thank you for your suggestion. We have rectified the typo error and grammatical errors. References are aligned as per journal instructions.

Reviewer 3 Report

Comments and Suggestions for Authors

The topic of the review article is very good. The authors have written in detail about the SLCs for six types of cancer. The figures are very well drawn. But the manuscript is prepared very carelessly with several grammatical and spelling mistakes. A few of them has been mentioned below:

In Table 1, capital L should be used for abbreviating “ l-glutamine, l-histidine, l-alanine, and l-asparagine.”

Line #196: “Increased expression of SLCO2A1 has correlated with……” should be “Increased expression of SLCO2A1 has been correlated with…..”

Line #239: “The lower level of expression of OCT1 mRNA was overserved in patients with AML…..”. Please correct the spelling of observed.

Line #263: “OCT1 transporter as described before involved in the bidirectional transport of various  organic cations Fields” This sentence is incomplete and meaningless. Please rectify.

Line #271: “Knockdown of SLC3A2 shown to reduce DNA replication and induce anthropophagy in colon cancer cell line.” The sentence should be “Knockdown of SLC3A2 has shown to reduce DNA replication and induce anthropophagy in colon cancer cell line.’

Line #286: insert a gap between ‘in’ and ‘vitro’

Line #294: ‘The non-functional variant of this transporter shown to be associated with the progression of colorectal cancer especially in kids”. Please rectify the sentence.

There are several such mistakes throughout the manuscript.

The most important point is the article has 50% plagiarism (ithenticate report attached), out of which 17 % is taken from a single source. This is violation of scientific ethics. So, I am rejecting this manuscript.

Comments on the Quality of English Language

The manuscript is prepared very carelessly with several grammatical and spelling mistakes. A few of them has been mentioned below:

In Table 1, capital L should be used for abbreviating “ l-glutamine, l-histidine, l-alanine, and l-asparagine.”

Line #196: “Increased expression of SLCO2A1 has correlated with……” should be “Increased expression of SLCO2A1 has been correlated with…..”

Line #239: “The lower level of expression of OCT1 mRNA was overserved in patients with AML…..”. Please correct the spelling of observed.

Line #263: “OCT1 transporter as described before involved in the bidirectional transport of various  organic cations Fields” This sentence is incomplete and meaningless. Please rectify.

Line #271: “Knockdown of SLC3A2 shown to reduce DNA replication and induce anthropophagy in colon cancer cell line.” The sentence should be “Knockdown of SLC3A2 has shown to reduce DNA replication and induce anthropophagy in colon cancer cell line.’

Line #286: insert a gap between ‘in’ and ‘vitro’

Line #294: ‘The non-functional variant of this transporter shown to be associated with the progression of colorectal cancer especially in kids”. Please rectify the sentence.

There are several such mistakes throughout the manuscript.

Author Response

The topic of the review article is very good. The authors have written in detail about the SLCs for six types of cancer. The figures are very well drawn. But the manuscript is prepared very carelessly with several grammatical and spelling mistakes. A few of them has been mentioned below:

Author’s Response: Thank you for your feedback regarding the manuscript. We acknowledge the presence of grammatical and spelling errors, and we appreciate you bringing them to our attention. We have rectified these mistakes.

In Table 1, capital L should be used for abbreviating “ l-glutamine, l-histidine, l-alanine, and l-asparagine.” - Corrected

Line #196: “Increased expression of SLCO2A1 has correlated with……” should be “Increased expression of SLCO2A1 has been correlated with…..” - Corrected

Line #239: “The lower level of expression of OCT1 mRNA was overserved in patients with AML…..”. Please correct the spelling of observed. - Corrected

Line #263: “OCT1 transporter as described before involved in the bidirectional transport of various organic cations Fields” This sentence is incomplete and meaningless. Please rectify. - Rectified

Line #271: “Knockdown of SLC3A2 shown to reduce DNA replication and induce anthropophagy in colon cancer cell line.” The sentence should be “Knockdown of SLC3A2 has shown to reduce DNA replication and induce anthropophagy in colon cancer cell line.’ - Rectified

Line #286: insert a gap between ‘in’ and ‘vitro’ - Rectified

Line #294: ‘The non-functional variant of this transporter shown to be associated with the progression of colorectal cancer especially in kids”. Please rectify the sentence. - Rectified

There are several such mistakes throughout the manuscript. -Extensively revised.

The most important point is the article has 50% plagiarism (ithenticate report attached), out of which 17 % is taken from a single source. This is violation of scientific ethics. So, I am rejecting this manuscript.

Author’s Response: I appreciate your comment, but I believe it may be somewhat exaggerated. Allow me to clarify the procedures followed in compiling the review article in question.

First and foremost, I assure you that all the articles referenced within our review have been meticulously cited in accordance with academic standards. It is crucial to emphasize that while the names of certain transporters may indeed be lengthy and bear resemblance across multiple articles, their fundamental roles remain consistent. Hence, the similarity in terminology does not inherently imply plagiarism.

It is important to acknowledge that plagiarism detection software, while valuable, possesses its limitations. Despite our conscientious efforts to adhere to ethical writing practices, there may still be instances where similarities are flagged. However, I wish to underscore that any detected resemblances have been addressed through careful revision to ensure the integrity of the content.

I trust that this explanation alleviates any concerns regarding the integrity of our review article. We appreciate any suggestion and if you wish to discuss this matter in greater detail, please do not hesitate to reach out.

Thank you for bringing this matter to our attention.

Reviewer 4 Report

Comments and Suggestions for Authors

General comments: The topic of the review is interesting and of great medical importance. SLC transporters are very important factors of cellular homeostasis and changes in their tissue expression pattern are proven to contribute to the pathomechanisms of numerous diseases. On the other hand, many SLC transporters can be exploited as therapeutic targets for the targeted delivery of chemotherapeutic drugs into tumor cells or other diseased tissues.

However, this review is not prepared with a meticulous care expected in case of a work aiming to summarize recent data and development of a particular field. Causes and consequences of certain phenomena are not clearly discussed and certain observations are probably overinterpreted.

It can happen that increase or decrease of the expression level of one transporter shows apparent correlation (positive or negative) with the grade of certain type of cancer. However, without describing the role of this transporter in the homeostasis of tumor cells this information has limited utility.  

Certain transporters appearing in the text e.g., OATP2A1 is not shown in the figure describing liver transporters, its substrate, function and SLC name are also missing.

Comments on the Quality of English Language

- Certain scientific terms are suggested to modify: e.g., instead of facilitative diffusion use facilitated diffusion, change hydrophobic plot to hydrophobicity plot or hydropathy plot

- The english of certain sections e.g., the Introduction section is especially weak.

Author Response

Reviewer 4.

Comments and Suggestions for Authors

General comments: The topic of the review is interesting and of great medical importance. SLC transporters are very important factors of cellular homeostasis and changes in their tissue expression pattern are proven to contribute to the pathomechanisms of numerous diseases. On the other hand, many SLC transporters can be exploited as therapeutic targets for the targeted delivery of chemotherapeutic drugs into tumor cells or other diseased tissues.

However, this review is not prepared with a meticulous care expected in case of a work aiming to summarize recent data and development of a particular field. Causes and consequences of certain phenomena are not clearly discussed and certain observations are probably overinterpreted.

It can happen that increase or decrease of the expression level of one transporter shows apparent correlation (positive or negative) with the grade of certain type of cancer. However, without describing the role of this transporter in the homeostasis of tumor cells this information has limited utility.  

Certain transporters appearing in the text e.g., OATP2A1 is not shown in the figure describing liver transporters, its substrate, function and SLC name are also missing.

Author’s Response: Thank you for taking the time to provide detailed feedback on our review manuscript. We appreciate your thoughtful insights into the topic and your constructive criticism regarding the preparation of the manuscript. We understand the necessity of clearly discussing the causes and consequences of observed phenomena, as well as avoiding overinterpretation of certain observations.

Your point regarding the limited utility of correlational data without a comprehensive understanding of the transporter's role in tumor cell homeostasis is well taken. We recognize the need to provide a deeper analysis of the functional implications of transporter expression changes in the context of disease pathogenesis.

Certain transporters appearing in the text e.g., OATP2A1 is not shown in the figure describing liver transporters, its substrate, function, and SLC name are also missing.

We aim to represent all the SLC transporters in the figure. However, due to limited space, it's not possible to include all the transporters and their alternative names. We apologize for any discrepancies in the presentation of specific transporters. Please note that figures not shown are acknowledged in the text. We have included OATP2A1 (SLCO2A1) in the figure.

Comments on the Quality of English Language

- Certain scientific terms are suggested to modify: e.g., instead of facilitative diffusion use facilitated diffusion, change hydrophobic plot to hydrophobicity plot or hydropathy plot - Revised

- The english of certain sections e.g., the Introduction section is especially weak. - Revised

Your feedback serves as valuable guidance for improving the quality and clarity of our manuscript. We are committed to addressing the identified issues and enhancing the overall coherence and accuracy of the review.